# *Pseudomonas aeruginosa* Biofilms

**DOI:** 10.3390/ijms21228671

**Published:** 2020-11-17

**Authors:** Minh Tam Tran Thi, David Wibowo, Bernd H.A. Rehm

**Affiliations:** Centre for Cell Factories and Biopolymers, Griffith Institute for Drug Discovery, Griffith University, Nathan QLD 4111, Australia; minhtam.tranthi@griffithuni.edu.au (M.T.T.T.); d.wibowo@griffith.edu.au (D.W.)

**Keywords:** *Pseudomonas aeruginosa*, biofilms, quorum sensing

## Abstract

*Pseudomonas aeruginosa* is an opportunistic human pathogen causing devastating acute and chronic infections in individuals with compromised immune systems. Its highly notorious persistence in clinical settings is attributed to its ability to form antibiotic-resistant biofilms. Biofilm is an architecture built mostly by autogenic extracellular polymeric substances which function as a scaffold to encase the bacteria together on surfaces, and to protect them from environmental stresses, impedes phagocytosis and thereby conferring the capacity for colonization and long-term persistence. Here we review the current knowledge on *P. aeruginosa* biofilms, its development stages, and molecular mechanisms of invasion and persistence conferred by biofilms. Explosive cell lysis within bacterial biofilm to produce essential communal materials, and interspecies biofilms of *P. aeruginosa* and commensal *Streptococcus* which impedes *P. aeruginosa* virulence and possibly improves disease conditions will also be discussed. Recent research on diagnostics of *P. aeruginosa* infections will be investigated. Finally, therapeutic strategies for the treatment of *P. aeruginosa* biofilms along with their advantages and limitations will be compiled.

## 1. Introduction

*Pseudomonas aeruginosa* is an ubiquitous Gram-negative bacterium that causes nosocomial infections, as well as fatal infections in immunocompromised individuals, such as patients with cancer, post-surgery, severe burns or infected by human immunodeficiency virus (HIV) [1,2,3]. In 2017, *P. aeruginosa* was recognised as one of the most life-threatening bacteria and listed as priority pathogen for Research and Development of new antibiotics by the World Health Organization [4]. Common antimicrobial agents like antibiotics frequently exhibit limited efficacy due to adaptability and high intrinsic antibiotic resistance of *P. aeruginosa*, thus increasing mortality [5]. Additionally, treatment of these infections is also hindered by the *P. aeruginosa* ability to form biofilms which protect them from surrounding environmental stresses, impedes phagocytosis and thereby confers capacity for colonization and long-term persistence [6]. Such ability is promoted by effective cell-to-cell communications within the microbial communities of *P. aeruginosa* known as quorum sensing. As a result, highly structured biofilms can be formed which is often identified in patients with chronic infections, such as chronic lung infection, chronic wound infection and chronic rhinosinusitis [7]. It has been estimated that biofilms have a substantial bearing on over 90% of chronic wound infections, resulting in poor wound healing. In the United States alone, approximately 6.5 million patients were affected by chronic wound infections, which resulted in high health-care burden and devastating economic consequences estimated at over US$25 billion annually [8]. It is important therefore to diagnose *P. aeruginosa* infections at an early stage before biofilm development which could enhance the susceptibility of *P. aeruginosa* towards antimicrobial treatments. However, the increasing incidence of acute and persisting infections worldwide also highlights the need to develop therapeutic strategies as an alternative to traditional antibiotics, expectedly to disarm and eradicate this Gram-negative bacterium.

This review highlights the *P. aeruginosa* biofilms starting from its composition, structure and development processes to the extraordinary capabilities of *P. aeruginosa* to invade host immune system and escape antibiotic treatments via biofilm-mediated resistance which is regulated mainly by quorum sensing. In the context of challenges facing *P. aeruginosa* devastating infections, recent diagnostics and therapeutic strategies will be discussed. 

## 2. *Pseudomonas aeruginosa* Biofilm

In nature, most bacteria can attach to different surfaces and form biofilms [9]. The biofilm is a complex aggregate of bacteria encased in a self-generated matrix of extracellular polymeric substances (EPS) and is one of the key strategies for the survival of species during unexpected changes of living conditions such as temperature fluctuation and nutrient availability [10]. Bacteria within a biofilm can escape host immune responses and resist antimicrobial treatments up to 1000 times more than their planktonic counterparts [11]. *P. aeruginosa* is a well-known biofilm former, which makes it an excellent model to study biofilm formation [12,13]. A resilient biofilm is a critical weapon for *P. aeruginosa* to compete, survive and dominate in the cystic fibrosis lung polymicrobial environment [14]. *P. aeruginosa* also effectively colonizes a variety of surfaces including medical materials (urinary catheters, implants, contact lenses, etc.) [12], and food industry equipment (mixing tanks, vats and tubing) [15]. Therefore, a greater understanding of the composition and structure of the biofilm, and the molecular mechanisms underlying the antimicrobial tolerance of bacteria growing within a biofilm, are vital for the design of effective strategies to manage, prevent and more importantly to eradicate biofilm-associated infections.

### 2.1. Biofilm Composition

The biofilm is a complex aggregate of bacteria encased in a self-generated matrix of extracellular polymeric substances (EPS) and is one of the key strategies for the survival of species against unexpected changes of living conditions such as temperature and nutrient availability [10,16,17].

It has been shown that *P. aeruginosa* biofilm matrix primarily encompasses polysaccharides, extracellular DNA (eDNA), proteins and lipids [12,18]. The matrix, which is responsible for more than 90% of biofilm biomass, acts as a scaffold for adhesion to biotic and abiotic surfaces and shelter for encased bacteria in harsh environmental conditions (antibiotics and host immune responses). It also provides a repertoire of public goods including essential nutrients, enzymes and cytosolic proteins for the biofilm community. The matrix also facilitates cell-to-cell communication [18,19,20]. 

The three exopolysaccharides, i.e., Psl, Pel and alginate, are tremendously involved in surface attachment, formation and the stability of biofilm architecture [12,21]. The roles of individual exopolysaccharides are discussed below. 

Psl is a neutral pentasaccharide typically comprising d-glucose, d-mannose and l-rhamnose moieties [22,23]. This exopolysaccharide is necessary for adhesion of sessile cells (cells attached to a surface) to surfaces and cell-to-cell interactions during biofilm initiation of both nonmucoid and mucoid strains [12,24,25]. Psl has the following characteristics: (i) Psl is beneficial for biofilm communities, but not for unattached populations; (ii) better growth of non-Psl producing cells was observed in mixed biofilm with Psl producing cells [21,26]; (iii) during biofilm growth, Psl positive populations dominate Psl negative populations; and (iv) Psl nonproducers are unable to exploit Psl producers [26]. In a mature biofilm, Psl is located in peripheries of the mushroom-like structure where it helps maintain structural stability [22]. Increased Psl expression is linked to induction of cell aggregates in a liquid culture which is a phenotype observed in CF patients’ sputum [27,28]. Psl functions as a signalling molecule to promote the production of c-di-GMP (bis-(3′-5′)-cyclic dimeric guanosine monophosphate) whose level, if elevated, results in thicker and more robust biofilms [28]. Additionally, Psl shields biofilm bacteria from antimicrobials [21] and neutrophil phagocytosis [29], making it an effective defence to achieve persistent infection. 

Pel is a cationic polysaccharide polymer of partially deacetylated *N*-acetyl-d-glucosamine and *N*-acetyl-d-galactosamine. Like Psl, Pel is an essential matrix component of biofilm in nonmucoid strains and is involved in the initiation of surface attachment, as well as maintenance of biofilm integrity [30,31]. Pel is responsible for the pellicle biofilm which is formed at the air-liquid interface of a static broth culture [32]. The synthesis of Psl and Pel are strain-specific and can be switched in response to surrounding conditions [33]. Pel promotes the tolerance to aminoglycoside antibiotics for biofilm-embedded bacteria [34]. Furthermore, Pel containing biofilms has been demonstrated to be refractory to the antibiotic colistin and less susceptible to killing mediated by neutrophils derived from human HL-60 cell lines [35]. Unlike Psl, Pel is not public goods and not available for Pel negative cells in both biofilm and unattached populations [26]. 

Alginate is predominately produced in the biofilm of mucoid *Pseudomonas* strains due to a mutation in *mucA22* allele. The mucoid phenotypes are found mostly in CF isolates, signifying the conversion from acute to chronic infection [36,37]. Alginate is a negatively charged acetylated polymer consisting of mannuronic acid and guluronic acid residues [38]. A wide range of important functions of alginate including biofilm maturation, protection from phagocytosis and opsonization, and decreased diffusion of antibiotics through the biofilm has been well-documented [18,39,40,41]. The ratios between mannuronic acid and guluronic acid influence the viscoelastic properties of biofilms which lead to impairment of cough clearance in the lung of CF patients infected with *P. aeruginosa* [42,43,44].

Cell lysis releases DNA into the environment, and this extracellular DNA (eDNA) is one of the crucial constituents of biofilms. Cell lysis can be caused by environmental stress such as the antimicrobial treatment via the endolytic activity of endolysin Lys which is encoded in the R- and F-pyocin gene cluster. This can occur in both early development of biofilms and the planktonic phase where rod-shaped bacteria rapidly turn into round cells resulting from structural damage of the cell wall and followed by lysis. The released eDNA, cytosolic proteins and particularly RNA are subsequently encapsulated into membrane vesicles (MVs) which are formed via membrane fragments originating from the lysed cells [45]. eDNA can also be localized on the surface and the stalk of the mushroom-like microcolonies [46,47]. eDNA is involved in various processes: (i) as a nutrient source for bacteria in the biofilm; (ii) supporting cellular organization and alignment via twitching motility; (iii) as a cation chelator that interacts with divalent cations (Mg^2+^ and Ca^2+^) on the outer membrane and subsequently activates the type VI secretion system which disseminates virulence factors within the host; (iv) the deposition of eDNA caused biofilm environment and infection sites to become acidic, limiting the penetration of antimicrobial agents; and (v) the presence of eDNA in *P. aeruginosa* biofilms can influence the inflammatory process activated by neutrophil [48,49,50,51]. 

It is noteworthy to mention the intracellular biopolymer, polyhydroxyalkanoate (PHA), that does not directly play a structural role in the biofilm matrix but is produced in cells within the biofilm. PHA, a carbon and energy storage polymer, has been implicated in stress tolerance as well as attachment to abiotic surfaces such as glass [52]. Within microaerophilic/anaerobic zones of the biofilm, PHA might serve as an electron sink to maintain energy-generating metabolic processes [6,16].

### 2.2. Biofilm Development

*P. aeruginosa* has been demonstrated to grow slowly as unattached cell aggregates under hypoxic and anoxic conditions, which are comparable to what has been observed in CF airways and chronic wounds [53]. Slow growth rates in the limited presence of oxygen are ascribed to antibiotic recalcitrance. Generally, biofilms of *P. aeruginosa* can be developed on abiotic surfaces, such as medical implants or industrial equipment. The biofilm development is divided into five distinct stages (Figure 1). Stage I: Bacterial cells adhere to a surface via support of cell appendages such as flagella and type IV pili [54,55]. The restricted flagellar movement has been implicated in mediating twitching motility and biosynthesis of exopolysaccharides required for surface association [56]. This adherence is reversible. A proteomic study with wild-type PAO1 concluded that the bacterial responses and biofilm formation are material specific. It is evident through records of the presence of specific bacterial proteins and their altered quantities when *P. aeruginosa* sense and react in response to a given surface [57]. Stage II: Bacterial cells undergo the switch from reversible to irreversible attachment. Stage III: Progressive propagation of attached bacteria into a more structured architecture, termed microcolonies. Stage IV: These microcolonies develop further into extensive three-dimensional mushroom-like structures, a hallmark of biofilm maturation. Stage V: In the center of the microcolony, matrix cavity is disrupted through cell autolysis for the liberation of dispersed cells [22] followed by the transition from sessile to planktonic growth mode for seeding of uncolonized spaces (Stage VI), which allows the biofilm cycle to repeat [58]. It was recently demonstrated that endonuclease EndA is required for dispersion of existing biofilm via eDNA degradation [59]. The structure of the formed biofilms is influenced by the swarming motility with flat biofilms resulting from highly motile bacteria, while mushroom-shaped biofilms are generated by cells with low motility, and that the motility rate was nutrient specific [60].

While it is well-perceived that biofilm cells are physiologically different from their planktonic counterparts and more recalcitrant to antimicrobial treatments [10], little is known about intermediate events between attached forms and the free-swimming lifestyle which generates highly virulent detached cells. It is suggested that the transition from detachment to planktonic growth involves a distinct stage of biofilm development (Stage VI) (Figure 1). These cells, in contrast to both planktonic and sessile cells, possess distinct physiology and represent the conversion from chronic to acute infections. They turn into a planktonic phenotype after remaining in a 2-h lag phase with decreased levels of pyoverdine and intracellular c-di-GMP [61,62]. Upregulation of virulence encoding genes and downregulation of iron uptake genes were observed in the dispersed population [10,63,64,65]. Both in vitro and in vivo experiments revealed that liberated cells are highly cytotoxic to macrophages, more sensitive to iron depletion and substantially virulent to nematode hosts relative to planktonic bacteria [63]. Notably, dispersed bacteria originating from biofilms treated with glycoside hydrolase rapidly disseminated and induced fatal septicaemia in a mouse chronic wound infection model [66].

### 2.3. Multispecies Biofilm

Generally, infections are not caused by monospecies alone but rather colonization of a complex polymicrobial community [67,68]. *P. aeruginosa* is often recognized as a co-colonizer along with other microbes such as *Staphylococcus aureus* (*S. aureus*), *Burkholderia cenocepacia* (*B. cenocepacia*) and *Streptococcus parasanguinis* (*S. parasanguinis*). For example, colonization of the biofilm-forming bacteria *P. aeruginosa* and *S. aureus* coinfects the lungs of CF patients and in diabetic and chronic wounds [69,70]. During co-infection, *P. aeruginosa* could sequester iron and nutrients through lysis of Gram-positive bacteria, including *S. aureus*, *Streptococcus pneumoniae* and *Bacillus anthracis* [67], as well as other Gram-negative bacteria *Burkholderia cepacia* [71]. 

In dual-species colonization containing *P. aeruginosa* and *S. aureus*, the presence of *S. aureus* derived peptidoglycan, *N*-acetylglucosamine (GlcNAc), induced *P. aeruginosa* to produce pyocyanin which functions as antimicrobials and toxins that could reduce the viability of *S. aureus* within the biofilm [68]. The transition from nonmucoid *P. aeruginosa*, which generate antimicrobial siderophores, rhamnolipids and 2-heptyl-4-hydroxyquinoline-*N*-oxide (HQNO), to mucoid phenotypes, which overproduce alginate, resulted in the decline of these exoproducts, leading to the cohabitant of *P. aeruginosa* and *S. aureus* [72]. *S. aureus* could also secrete extracellular adhesin known as staphylococcal protein A (SpA) which bound to Psl and type IV pilli on the *P. aeruginosa* cell surface, resulted in the inhibition of both biofilm formation of *P. aeruginosa* and phagocytotic activity of neutrophil towards *P. aeruginosa* [73]. *P. aeruginosa* has been demonstrated to outcompete *S. aureus* in the dual-species community by producing diguanylate cyclase, SiaD, which is activated by Psl, during the early stage of biofilm formation [74]. 

Similarly, synergistic interactions between *P. aeruginosa* and *B. cenocepacia* have been observed [75]. In planktonic co-cultures, *P. aeruginosa* predominated due to the production of secondary metabolites which inhibited the growth of *B. cenocepacia*. Moreover, co-existence in *B. cenocepacia* biofilm promoted biofilm biomass of *P. aeruginosa.* Co-infection of the two species was found to advance lung damage in a mouse model [75]. 

Although *P. aeruginosa* remains dominant in mixed-species biofilms by producing antimicrobial compounds which modulate the growth of other microorganisms, its pathogenesis was shown to be inhibited by oral streptococci strains during co-infection, resulting in improved CF lung conditions [76]. The oral commensal streptococci outcompeted *P. aeruginosa* by the production of hydrogen peroxide in the presence of nitrite [77]. Furthermore, it was shown that oral commensal *S. parasanguinis* could effectively exploit the exopolysaccharide alginate produced by *P. aeruginosa* CF isolate FRD1 strain to promote its biofilm in vitro through the mediation of the streptococcal surface adhesin BapA1. However, either adhesin BapA1 or Fap1 was adequate to reduce the colonization of alginate producing *P. aeruginosa* in *Drosophila melanogaster* [78].

## 3. Quorum Sensing in Biofilm Development

The development of *P. aeruginosa* biofilms requires population-wide coordination of individual cells within bacterial communities [79]. *P. aeruginosa* uses multiple interconnected signal transduction pathways known as quorum sensing (QS), enabling the bacteria to communicate between the individual cells and ultimately orchestrate collective behaviour which is essential for the adaptation and survival of whole communities. *P. aeruginosa* enters into the QS mode in response to changes in cell density and environmental cues or stresses [80]. QS involves the production, secretion and accumulation of signalling molecules called autoinducers (AI) whose specificity and concentration are sensed by transcriptional regulators [81], resulting in the expressions of specific sets of genes on a population-wide scale. In addition to biofilm development, QS has been linked to the regulation of other physiological processes, including virulence-factor production, stress tolerance, metabolic adjustment and host-microbe interactions [6]. Thus, understanding and controlling these chemical communication systems could lead to new targets for alternative or complementary treatments to conventional antimicrobials and antibiotics.

There are four distinct pathways in the QS circuits of *P. aeruginosa*, namely Las, Rhl, PQS and IQS that intracellularly produces their cognate AI molecules, i.e., *N*-3-oxo-dodecanoyl-ʟ-homoserine lactone (3O-C_12_-HSL), *N*-butyryl-ʟ-homoserine lactone (C_4_-HSL), 2-heptyl-3-hydroxy-4-quinolone (PQS) and 2-(2-hydroxyphenyl)-thiazole-4-carbaldehyde (IQS), respectively (Figure 2). These QS circuits are organized in a hierarchy with the Las system at the top of the cascade [82]. Both Las and Rhl systems are triggered by an increased cell density at the preliminary exponential growth phase, whereas PQS and IQS systems are activated at late exponential growth phase [83] especially under iron limitation [84] and phosphate starvation conditions [85], respectively. The synthesized AIs undergo membrane trafficking directed to outside then inside of the cells, presumably mediated by free diffusion, efflux pumps or outer membrane vesicles [86]. The trafficked 3O-C_12_-HSL is then bound to the regulator protein LasR, and the formed complex activates *lasI* synthase gene, leading to the autoinduction feed-forward loop [87]. The LasR–3O-C_12_-HSL also induces the expression of *rhlR* and *rhlI* genes as well as the *pqsR* and *pqsABCDH* genes which encode the Rhl [82] and the PQS [88] systems, respectively. Similar to the Las system, the RhlR–C_4_-HSL complex induces *rhlI* gene expression that activates the second autoinduction feed-forward loop [89]. In the PQS system, the PqsR–PQS complex activates *pqsABCDH* genes as well as feeds back to induce *rhlRI* gene expression [90]. The expression of both *pqsR* and *pqsABCDH* genes can be inhibited by RhlR, which has been suggested as a way to control the correct ratio between 3-oxo-C_12_-HSL and C_4_-HSL, hence controlling the activation of PQS pathway [91]. The identification of IQS system was relatively new as compared to the other QS systems, In the IQS system, the identity of the transcriptional regulator is still unknown although its binding to the IQS has been found to activate the *pqsR* gene [92,93]. In addition, the IQS molecule was proposed to be enzymatically produced from the proteins encoded by *ambBCDE* genes [92]. However, the recent identification of the IQS molecule (an aeruginaldehyde) revealed that it is a byproduct of the siderophore pyochelin biosynthesis [94,95]. On the other hand, *ambBCDE* genes encode for proteins involved in the biosynthesis of the anti-metabolite L-2-amino-4-methoxy-trans-3-butenoic acid (AMB) [96].

The Las, Rhl and PQS systems in the QS network of *P. aeruginosa* play important roles in the production of the functional elements that have an impact on biofilm development (Figure 2). These include rhamnolipid [97,98], pyoverdine [99], pyocyanin [100,101], Pel polysaccharides [102], and lectins [103,104]. Rhamnolipid is a rhamnose-containing glycolipidic compound (i.e., biosurfactant) that functions to preserve the pores and channels between microcolonies, enabling the passage of liquid and nutrients within mature biofilms. Pyoverdine can sequester iron in the environment and delivery it to the cell which is an essential component for biofilm development. In an environment where iron is limited, twitching motility is more favoured than sessile growth and thus inhibiting biofilm formation [105]. Pyocyanin is a secondary metabolite with a cytotoxicity effect, thereby inducing cell lysis and releasing the cells’ DNA to extracellular space (i.e., eDNA–one of the biofilm components). Pyocyanin can bind to the eDNA and causes an increase in solution viscosity, thus also increasing the physicochemical interactions between biofilm matrices and the surrounding environment as well as promoting cellular aggregation. Pel polysaccharides can also interact with eDNA through cationic-anionic interactions within the biofilm matrix, strengthening the biofilm structure. Lectins are soluble proteins located in the outer membrane which consist of two forms, i.e., LecA (that binds to galactose and its derivatives) and LecB (that binds to fucose, mannose and mannose-containing oligosaccharides). Such adhesive properties of lectins facilitate the retention of both cells and exopolysaccharides in a growing biofilm, contributing to the biofilm structure, as well as adhesion to biological surfaces such as epithelium and mucosa. Collectively, such molecular and cellular interactions in combination with other polymeric substances lead to the establishment of a robust and mature biofilm.

## 4. Diagnostics

Rapid and accurate diagnostics for *P. aeruginosa* infections, especially at the early stages, are particularly important to ensure effective treatments and to prevent the conversion to devastating chronic infections. Although the conventional microbiological culture of patient sputum is a routine procedure in most of the laboratories, there are currently many innovative technologies that have been developed to improve speed, accuracy and specificity of detection methods, and thereby advancing management and surveillance of diseases. Here, the development of diagnostic tests and their advantages and disadvantages will be discussed. 

### 4.1. Conventional Microbiological Culture

Misidentification of *P. aeruginosa* infections is not uncommon and is an increasing concern in the clinical environment as the bacteria exhibit diverse morphologies such as small colony variants, rough small colony variants, wrinkle variants, etc. As the diseases progress to chronic infections with biofilm formation, the isolated bacteria display distinct phenotype—cell aggregates which form “sticky” colonies [106]. Importantly, biofilm bacteria from long-term chronic infections in CF lung exhibit slow growth or uncultivability, which make biofilm infections difficult to be identified by conventional microbial culture of clinical specimens and biochemical reactions [107,108]. The use of automated identification systems including BD Phoenix [107], the bioMérieux Vitek 2 [109] and MicroScan WalkAway [110] for bacterial detection and antimicrobial susceptibility testing (AST) have been broadly used in several clinical settings. Various readouts such as turbidimetric, kinetic, colorimetric and fluorescent signals resulting from microbial growth, a variety of enzymatic based reactions and broth microdilution tests are measured to detect the bacterial presence and antimicrobial susceptibility [111,112,113]. Nonetheless, it has been reported that these expensive systems lack specificity and accuracy, and required highly skilled personnel, regular software upgrade and frequent reference database update [109,114]. 

### 4.2. Molecular Biology Methods

Polymerase chain reaction (PCR) is one of the most common methods utilized for the identification of *P. aeruginosa*. It has been designed to target several specific genes such as the 16S rRNA, *oprI, oprL*, *algD*, *gyrB*, *toxA*, *ecfX*, *ETA,* and *fliC*. Further examination of the specificity of *P. aeruginosa* genes showed that *ecfX*, *oprL* and *gyrB* have high sensitivity and specificity [115,116]. Multiplex PCR targeting more than one gene of interest in a single reaction may be an option to effectively address false positive and false negative results obtained using conventional PCR. This method, albeit, faces the main disadvantage of designing primers with high specificity in the abundant presence of gene targets [117]. Real-time fluorescence-based quantitative PCR (RT qPCR) has been established to identify pathogens accurately and specifically. Numerous studies have developed RT qPCR assay to detect *P. aeruginosa* in CF patients with shorter turnaround time [116,118,119]. Another approach is polymerase spiral reaction (PSR) which has allowed rapid identification of *P. aeruginosa* in the sputum of intensive care unit (ICU) patients within 60 min under isothermal conditions with 10-fold higher sensitivity compared to conventional PCR [120]. 

### 4.3. Mass Spectrometry

Matrixed-assisted laser desorption/ionisation time-of-flight mass spectrometry (MALDI-TOF MS) has gained popularity for speedy and reliable detection of *P. aeruginosa* in microbiology practice with high throughput capabilities at low costs [121]. It generates ribosomal protein-based peptide mass fingerprint (PMF) profiles of investigated microorganisms which are compared to reference PMF database to identify and characterize bacteria in clinical samples [122,123]. Microbial identification by MALDI-TOF MS was not affected by culture conditions such as culture media and culture duration [124,125]. Recent applications of this technique include characterization of species and identification of resistance or virulence biomarkers of multidrug-resistant pathogens [126,127]. Additionally, MALDI-TOF MS profiling has been demonstrated to be capable of differentiating different biofilm stages and capturing phenotypic changes during biofilm development. This technique may be a valuable tool for clinical biofilm infection detection, particularly in early stages, and thus improvement of patient outcomes [128]. 

### 4.4. Nanoparticle Biosensor

Quorum sensing of *P. aeruginosa* produces specific molecules, including pyocyanin and LasA protease, that have been subjected to analyses for detection of *P. aeruginosa*. Traditional methods to identify pyocyanin are based on ultraviolet-visible (UV-Vis) spectrophotometry that detects absorbance peaks at both 382 and 521 nm, and liquid chromatography-mass spectrometry (LC-MS) that measure molecular mass at 211 g/mol [129]. On the other hand, LasA protease can be identified using sodium dodecyl sulfate-polyacrylamide gel electrophoresis [130]. However, those methods are time-consuming as they require the target molecules/proteins to be purified from bacterial cultures. Recently, nanoparticles have been developed to allow rapid identification of pyocyanin [131] and LasA protease [132] with high sensitivity even in the presence of other species within cell cultures. Gold nanoparticles (Au NPs) deposited onto indium tin oxide electrodes was able to enhance the detection of pyocyanin at a concentration as low as 40 µM as compared to the minimum 80 µM of pyocyanin detected using unmodified electrodes [131]. Encapsulating the Au NPs with a thin layer of polyaniline hydrochloride (PANI) forming PANI/Au core-shell NPs increased the sensitivity of the PANI/Au NPs-modified electrodes towards pyocyanin down to 36 µM, which could accelerate the diagnostic process, achieving effective treatments and prevention of chronic infections [131]. Magnetic NPs have also been developed as biosensors toward LasA protease by functionalizing the NPs with (glycine)_3_ peptides and then linking them onto a gold-coated paper substrate [132]. LasA protease in the cell cultures could cleave the peptides from NPs, and the cleaved NPs could attract to external magnetic forces revealing the golden colour of the sensor surface which could then be detected colorimetrically [132]. Such simple biochip allowed rapid identification of *P. aeruginosa* in clinical samples such as sputum, ear and wound in less than one minute with a detection limit of 10^2^ cfu/mL [132], which could expectedly become a useful point-of-care diagnostic device. 

## 5. Therapeutic Strategies

Therapeutic management of *P. aeruginosa* infections poses unique challenges for the clinical use of conventional antimicrobials. This bacterium displays multiple drug tolerance mechanisms which can be categorized into intrinsic, acquired and adaptive mechanisms. Biofilm formation as an adaptive mechanism is considered as the key virulence factor enhancing the survival of exposure to antibiotics and initiating chronic infections [133]. As the development and dispersal of biofilm are regulated by a multifactorial process entailing quorum sensing systems, exopolysaccharides and c-di-GMP [58], biofilm remediation strategies are required to target different constituents of biofilm matrix and biofilm-residing cells [134]. Moreover, it is critical to consider the interaction between the host immune system and the infectious agents when treating biofilm-related infections [135,136]. Importantly, clinical chronic infections are frequently diagnosed with co-infections of multi-species, which profoundly worsen the patient outcomes as compared to mono-infections. This has greatly challenged biofilm therapeutic strategies aiming at disrupting biofilm community of mono-species [137,138]. Efforts have been developed to tackle these challenges by targeting biofilm components, inducing biofilm dispersal, inhibiting quorum sensing and targeting iron metabolism using various antimicrobial agents, including antimicrobial peptides, biofilm-degrading enzymes, quorum sensing inhibitor, and iron chelator. We provided a summary of current therapeutic strategies targeting *P. aeruginosa* biofilms including advantages and limitation of individual approaches (Table 1). 

### 5.1. Nanoparticles

Nanoparticles (NPs) have been used to assist in the delivery of such antimicrobial agents to the sites of infection such as penetration of antibiotics into biofilms. There are other promising delivery systems including liposomes, solid lipid NPs and polymeric NPs whose in vivo efficiency, potential cytotoxicity and interactions between host immunity and NPs, still await further investigation [177,178,179]. In addition to its carrier functionality, NPs have been used as an antimicrobial agent. For example, silver NPs were shown to have antimicrobial action with minimal cytotoxicity against clinical *P. aeruginosa* strains from nosocomial infections which are resistant to certain antibiotics [155]. Furthermore, polyphosphoester NPs co-loaded with silver acetate and minocycline substantially increased the susceptibility of *P. aeruginosa* [156]. Conjugation of azlocillin antibiotic onto silver NPs resulted in significantly promoted antibacterial efficacy compared to either NPs alone, or azlocillin alone, or co-administration of NPs and azlocillin [180]. 

### 5.2. Targeting EPS Component and Structure

Therapeutic approaches targeting components of biofilm matrices are an option to tackle *P. aeruginosa* infections. There are several types of enzymes employed to degrade biofilm matrix: alginate lyases [181], glucanohydrolases (e.g., dextranase and mutanase) [182], glycoside hydrolase (e.g., PelAh and PslGH) [35], and deoxyribonucleases (e.g., DNase I and DNase1L2) [183,184]. Furthermore, therapies involving a combination between biofilm matrix-degrading enzymes and antibiotics can facilitate biofilm dispersal, improve drug penetration, and therefore maximizing efficacy against established biofilms [160,185]. 

Alginate inhibitor, such as thiol-benzo-triazolo-quinazolinone, has been demonstrated to interfere with the binding of Alg44 to c-di-GMP. This interference resulted in a declined alginate production in *P. aeruginosa*, which could potentially limit mucoid conversion and reduce a complication in CF patients [186]. Alginate oligomer (i.e., OligoG) inhibited biofilm formation by inducing disruption of mucoid biofilm through interaction with both EPS and eDNA [161]. 

### 5.3. Immunotherapies

Bispecific monoclonal antibodies have been investigated as a promising agent to manage the bacterial infection. Anti-Psl, anti-PcrV (Type III Secretion System protein) and BiS4αPa (known as MEDI3902) have demonstrated their potential effectiveness towards both management and prevention of infection [187,188]. Monoclonal antibodies targeting a family of bacterial DNA-binding proteins (DNABII), a common component of biofilm which supports the structural stability, have been proved to be promising therapeutic candidates against *P. aeruginosa* biofilm in a mouse lung infection model [189]. 

### 5.4. Induction of Biofilm Dispersal

The signal molecule nitric oxide (NO) has been reported to induce biofilm dispersal through modification of intracellular c-di-GMP levels. However, biofilm dispersal was not enhanced by multiples treatments with NO due to the presence of flavohemoglobin in *P. aeruginosa* biofilms. To overcome this, imidazole was utilized to inhibit NO dioxygenase activity of flavohemoglobin produced in NO-pretreated biofilms and restored dispersal activity [162]. Furthermore, low dose NO gas can be used as an adjunctive strategy to potentiate antimicrobial efficacy by dispersing biofilm and exposing dispersed bacteria to antibiotic killing in ex vivo and clinical studies [190].

### 5.5. Inhibition of Quorum Sensing

Quorum sensing is an attractive target for inhibition and eradication of biofilm. Naturally occurring carotenoid zeaxanthin was found to inhibit biofilm and target LasIR and Rh1IR QS systems of wild-type PAO1 [191]. The plant flavonoid quercetin has been known for its their pharmacological effects such as reducing the production of pyocyanin and inhibiting the biofilm formation in *P. aeruginosa* [192]. A benzamide-benzimidazole compound, M64, restricted biofilm development and improved antibiofilm properties of meropenem and tobramycin achieved by inhibiting the controller of QS system, MvfR [193]. Another quorum quenching molecule, *P. aeruginosa* periplasmic enzyme PvdQ acylase, was proved to be a hydrolyzer of *N*-acyl homoserine lactone (AHL), thereby attenuating virulence and alleviating infections in vitro and in a mouse pulmonary infection model [194]. Notably, terrein, a bioactive fungal metabolite isolated from *Aspergillus terreus*, has been revealed for the first time as an inhibitor with dual activities against both QS system and c-di-GMP without affecting bacterial viability, thus limiting the development of drug resistance [164]. A recent study identified a quorum quenching enzyme AqdC which decreased the production of alkylquinolone and pyocyanin while increasing levels of elastase. Interestingly, the co-presence of *Rhodococcus erythropolis* isolated QsdA and AqdC resulted in the decline of *N*-acylhomoserine lactone, rhamnolipid as well as elastase levels, suggesting that targeting a complex quorum sensing network may require more than a single enzyme. Indeed, the presence of both enzymes improved the survival of *Caenorhabditis elegans* upon *P. aeruginosa* exposure [163]. 

### 5.6. Targeting Iron Metabolism

Iron is known to play vital roles in a range of cellular processes including bacterial growth, DNA replication, biofilm formation and infection establishment [195,196]. Evidently, the enhanced concentration of sputum iron was observed in the lung of CF patients compared to healthy controls [197]. *P. aeruginosa* acquires extracellular iron via iron uptake systems such as the siderophores pyocyanin and pyochelin. Hence, targeting iron metabolism using iron analogues and chelators could be promising strategies to combat *P. aeruginosa* infections. Gallium, structurally similar to iron, was utilized as an iron substitute to impede iron uptake, disrupt iron-dependent pathways, influence bacterial viability and disturb biofilm formation [166]. Combination of tobramycin and FDA-approved iron chelation compounds, i.e., deferoxamine and deferasirox, is another effective approach to disperse pre-existing biofilms, enhance bactericidal activity and prevent biofilm development [167]. Treatment using novel deferoxamine conjugated gallium complexes alone exhibited planktonic killing and biofilm formation disturbance, and when co-administered the complexes with gentamicin showed substantially enhanced antibiofilm actions [198].

### 5.7. Photodynamic Therapy

Photodynamic therapy (PDT) has been demonstrated as a promising approach to tackle bacterial infections in both planktonic and biofilm lifestyles. Its antimicrobial effect relies on the activation of non-toxic photosensitizers (PS) upon exposing to harmless visible light of appropriate spectra in the presence of oxygen to produce cytotoxic reactive oxygen species (ROS) on site [199,200]. These oxygen species act on multiple targets including biofilm matrix, cell membranes and other cellular organelles as well as macromolecules such as lipids, protein and nucleic acids [201]. Unlike conventional antimicrobials, the mechanism of PDT involves a distinct molecular pathway, hence unlikely inducing resistances [202]. As the efficacy of PDT mainly depends on the PS uptake by bacterial cells, different PS molecules have been employed to improve the cellular internalization [172,203]. Treating *P. aeruginosa* biofilms with the PS, tetracationic porphyrin [5,10,15,20-tetrakis(1-methylpyridinium-4-yl)porphyrin tetra-iodide, Tetra-Py^+^-Me], has resulted in a significant decline of EPS content, suggesting that the EPS may be the target of photoinactivation [204]. GD11, a member of a boron-dipyrrolemethene (BODIPY) dye family, has also been developed as PS against wild-type *P. aeruginosa* PAO1 biofilms [170]. After the irradiation, a substantial cell lethality was observed with minimal impact on biofilm structure and without recurrence of microbial growth after 24h [170]. A glycosylated photosensitizer pGEMA-I derived from BODIPY has recently been developed to selectively bind to bacteria over human cells [205]. This improved selectivity has led to high bactericidal activity, effective biofilm suppression and reduction of side effects [205]. A recent study has revealed that curcumin mediated PDT was capable of efficiently decreasing viability of both planktonic and biofilm bacteria, and successfully repressing EPS production via inhibition of QS system [206]. Moreover, drug delivery systems such as nanoparticles are also utilized to enhance the stability and the diffusion of the PS into the biofilms. For example, the PS toluidine blue (TB) loaded in mesoporous silica nanoparticles was capable of considerably reducing the cell viability and biofilm biomass as compared to free TB [172].

### 5.8. Photothermal Therapy

Photothermal therapy (PTT) has emerged as an attractive therapeutic modality for combating antibiotic-resistant bacterial infections. PTT involves the use of the near-infrared (NIR) light in a wavelength ranging from 700 to 1100 nm for irradiation of nanomaterials to generate localized heat which causes irreversible damage to nearby bacterial cells [207]. Various photothermal nanomaterials, including carbon-based nanocomposites [208,209,210], gold and silver nanoparticles [211,212] and conducting polymers [207,213], have been optimized for PTT. For example, gold nanorods (AuNR) surface engineered with phospholipid-polyethylene (PEG) have been developed and utilized against *P. aeruginosa* suspension and biofilm cultures [173]. Upon continuous or pulse activation of laser, severe cell membrane destruction of planktonic cells and marked lethality of the biofilm population were observed [173]. Thiolated graphene (TG) sheet has also been developed for photothermal therapy by functionalizing with S-nitrosothiols, a heat-sensitive NO donor, along with boronic acid, which facilitates cell attachment due to its dense hydroxyl groups, to obtain a multifunctional photothermal agent [175]. Application of this agent with single NIR has led to the simultaneous release of NO and localized heat, both of which are required for effective ablation of bacteria and biofilms in vitro and in vivo with fewer side effects [175]. Recently, bacteriophage has been developed for treatment against antibiotic-resistant bacterial infections due to its high affinity toward bacteria. To overcome the cytotoxicity limitation of bacteriophage-based antibacterial approach, the bacteriophage was conjugated with AuNR [174]. This bioconjugate, phage-AuNR, has shown rapid killing of both planktonic and biofilm cells with minimal effect on mammalian epithelial cells and successful inhibition of phage replication. In addition, this strategy has offered improved specificity towards targeted pathogens via aggregation of phage-AuNR on bacteria, thus it is possibly used for diagnosis of bacterial infections as well [174]. 

## 6. Conclusions

Antimicrobial treatments of *P. aeruginosa* infections are still challenging which is mostly due to the ability of *P. aeruginosa* to form dense and persistent biofilms. Biofilms of *P. aeruginosa* are composed of polysaccharides (Pel, Psl and alginate) and extracellular DNA that play critical roles in protecting the bacterial communities from exogenous stresses caused by antimicrobial agents. This extraordinary ability of *P. aeruginosa* to form biofilms is promoted by sophisticated cell communication system called quorum sensing that function in a hierarchical manner and are actively inducible upon an increase in cell density or via limitation of nutrients (e.g., iron and phosphate). The complexity of antimicrobial treatments for *P. aeruginosa* biofilm is compounded when multiple species are also involved in forming polymicrobial communities within the biofilms. Molecular diagnosis methods have been developed to detect and identify *P. aeruginosa*. These methods are heavily relied on the *P. aeruginosa* characteristics in terms of its genetic, microbial physiology and biochemical markers that can be characterized using, for example, polymerase chain reaction, cell cultures for antibiotic-resistance profiling, and nanoparticle biosensor, respectively. Despite the diagnosis advancement, major challenges include the enrichment of bacterial cells and biochemical markers for detection and the lack of biomarker database, as well as the high cost and/or time-consuming. In the perspective of therapeutic strategies, more targeted approaches for the treatment of *P. aeruginosa* infections have been developed in the past years in efforts to target different components governing biofilm structure, biofilm dispersal, quorum sensing, and iron metabolism. Although promising results have been demonstrated, translation to clinical trials seems to be challenging due to the variation in biofilm lifestyle, composition and phenotypes which depend on various parameters such as nutrition conditions and the presence of other bacterial species. Future studies are required on the development of more advance techniques, aiming at providing high-throughput and specific diagnosis at an early stage of *P. aeruginosa* growth prior to biofilm development. On the other hand, more thorough understanding on genetic pathways that are responsible for all biofilm lifestyle cycles of *P. aeruginosa* warrants further investigation, which could provide informed decision to design therapeutic strategies that could impair the bacterial attachment and biofilm maturation capabilities.

## Figures and Tables

**Figure 1 ijms-21-08671-f001:**
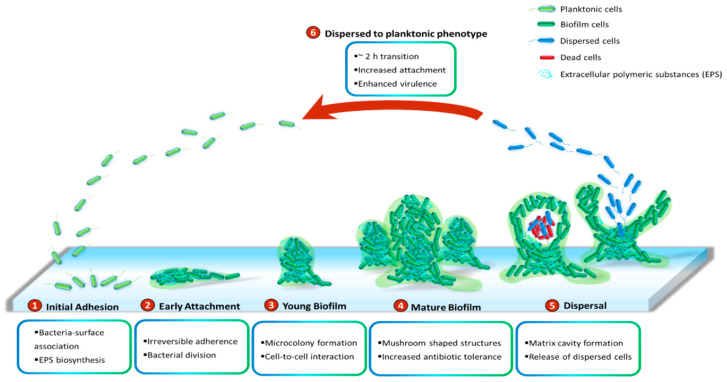
Cycle of *P. aeruginosa* biofilm development. The development cycle is divided into six stages. Initially, the bacteria associate with the surface and produce extracellular polymeric substances (EPS) including proteins, polysaccharides, lipids and eDNA. Next, cell division and the transition of reversible attachment into irreversible take place. The following steps are the formation of microcolonies and the further development of these microcolonies into mushroom-shaped structures. Cell-to-cell interaction and production of virulence factors play essential roles in maturation and robustness of biofilms. Matrix cavity is then formed in the centre of microcolony via cell autolysis to disrupt the matrix for the liberation of the dispersed population. Finally, the released cells undergo an approximately 2 h transition into planktonic phenotypes which subsequently occupy uncolonized spaces.

**Figure 2 ijms-21-08671-f002:**
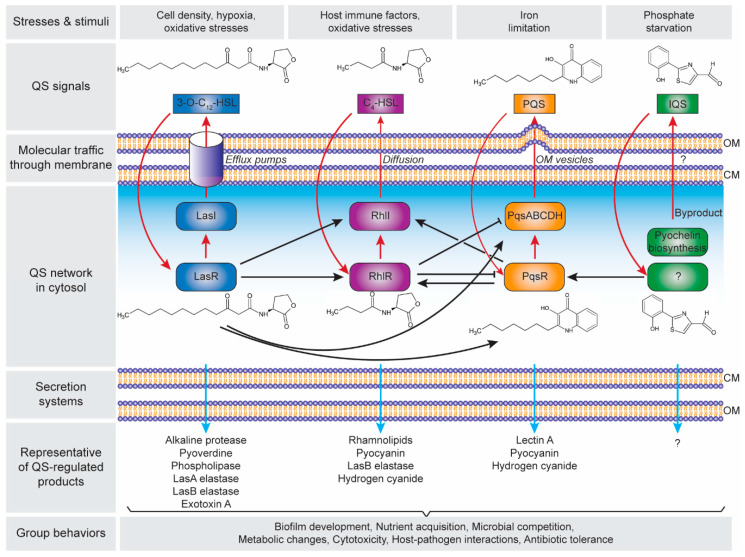
Hierarchical quorum-sensing (QS) network in *Pseudomonas aeruginosa*. The four QS pathways are activated in response to the cell density and environmental stimuli, with four autoinducer synthases including LasI, RhlI, PqsABCDH and AmbBCDE that produce *N*-3-oxo-dodecanoyl-ʟ-homoserine lactone (3O-C_12_-HSL), *N*-butyryl-ʟ-homoserine lactone (C_4_-HSL), 2-heptyl-3-hydroxy-4-quinolone (PQS) and 2-(2-hydroxyphenyl)-thiazole-4-carbaldehyde (IQS), respectively. Note: the autoinduction is depicted in red arrows; the receptor for IQS is still unknown. The QS products are secreted through the cell membrane, that control the group behaviours and essential for the adaptation, survival and pathogenicity of *P. aeruginosa*. Abbreviation: CM, cytoplasmic membrane; OM, outer membrane.

**Table 1 ijms-21-08671-t001:** Summary of therapeutic strategies against *P. aeruginosa* infections.

Therapeutic Approach	Activity	Advantages	Limitation	References
**Antimicrobial peptides**	Antibacterial Antibiofilm Immunological modulator	Low cytotoxicity Combined treatment possibility Low resistance	Sensitive to salt, serum and pH Susceptible to host proteolysis Expensive production	[139,140,141,142,143,144]
**Antibiotics**	Antibacterial	Inhaled antibiotic class Broad-spectrum Safety Improvement of lung function in CF patients	Resistance development	[145]
**Lectin inhibitor**	Antibiofilm	High stability Low resistance	No in vivo data Toxicity Narrow spectrum	[146,147,148]
**Bacteriophages**	Antibacterial	Delivery at the infection site High specificity Fewer side effects Easy administration	Poor stability Undesired cytotoxicity Resistance development Insufficient pharmacokinetics and pharmacodynamics data	[149,150,151,152,153]
**Natural products**	Antibacterial Antibiofilm Quorum sensing modulator	Broad-spectrum Multiple mechanisms of action	Cytotoxicity Resistance development Limited penetration into biofilm Limited killing effects on slow-growing bacteria Availability and Complex extraction and isolation	[154]
**Nanoparticles**	Antibacterial Antibiofilm	Broad-spectrum Combination with antibiotics/therapeutic agents Small size, thus direct delivery to targets	Cytotoxicity Host metabolism of nanoparticles	[155,156]
**Nanocarriers (Liposomes, solid lipid and polymeric)**	Drug delivery	Protection of therapeutic agents from inactivation and degradation by bacterial and host system Enhancement of efficacy Penetrability into the biofilm matrix	Cytotoxicity Host metabolism of nanoparticles	[157,158,159]
**EPS inhibitors**	Anti-EPS	Biofilm matrix degradation Limited/no effect on bacterial viability Low risk of resistance development Augmentation of antibiotic efficacy to clear the infection	Incomplete biofilm matrix disruption Cytotoxicity	[35,160,161]
**Biofilm dispersers**	Dispersal induction	Augmentation of antibiotic efficacy to clear the infection Promotion of self-disassembly Low risk of resistance development	Release of harmful dispersed cells for re-colonization and lethal septic event Cytotoxicity	[66,162]
**QS inhibitors**	Biofilm prevention Biofilm inhibition	Reduction of virulence factors No effect on bacterial viability	Narrow spectrum Unwanted effect on bacteria	[163,164,165]
**Iron chelator**	Interference with iron metabolism	Bactericidal activity Biofilm prevention Low risk of resistance development	Cytotoxicity	[166,167]
**Photodynamic therapy**	Antibacterial Antibiofilm	No resistance development Improved selectivity No photocytotoxicity	Potential side effects (e.g., burns, redness swelling of treated skin)	[168,169,170,171,172]
**Photothermal therapy**	Antibacterial Antibiofilm	No resistance development Improved selectivity Negligible cytotoxicity	Photothermal ablation of host tissues	[173,174,175,176]

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
