# Peer review of "Pseudomonas aeruginosa Biofilms"

_ijms, 2020, doi:10.3390/ijms21228671_

Round 1

Reviewer 1 Report

The current review entitled “Pseudomonas aeruginosa biofilms” discusses the opportunistic human pathogen Pseudomonas aeruginosa which can cause infections in individuals with compromised immune systems. Being a nosocomial pathogen it has the ability to form antibiotic-resistant biofilms specifically in CF of lungs. This review discusses the various aspects of Pseudomonas aeruginosa biofilms during the development stages, molecular mechanisms for invasion and persistence, diagnostics, and eradication therapies utilized.

          In the section discussing the biofilm composition, I was curious to know the nature of Psl exopolysaccharide, whether it is positively or negatively charged? Pel and Alginate are mentioned as cationic and anionic respectively.

          In the quorum sensing section, check ref- (Lee et al. 2013), appears twice as ref no. 93 and 95 and as (Lee et al. 2013) format.

          Table 1 describes the summary of therapeutic strategies against P. aeruginosa infections which need reformatting. The current format made it difficult to comprehend initially.

I found that the therapeutic approaches section doesn’t discuss, examples of antimicrobial peptides, antibiotics, lectin inhibitors, bacteriophages, and natural products except in table 1 and only focuses on the remaining like nanoparticles, targeting EPS components, immunotherapies, biofilm dispersal, inhibition of QS, and iron chelation. Any specific reason for this distribution should be commented on.

In the introduction, authors have mentioned: “the increasing incidence of acute and persisting infections worldwide also highlights the need to develop therapeutic strategies as an alternative to traditional antibiotics, expectedly to disarm and eradicate this Gram-negative bacterium.”

Although, the above mentioned therapeutic approaches discussed are alternative, can antimicrobial photodynamic inactivation/therapy (aPDI/T) count as an alternative therapy?

For further reading please refer to-

General aPDI reading- 1. Critical Reviews in Microbiology, 2018, 44:5, 571-589, (DOI:10.1080/1040841X.2018.1467876).

Considering that the structure of biofilms is mainly composed of exopolysaccharides, (depending on their charges as described in the biofilm composition section), the aPDT therapeutic method targets cell wall structures of bacteria by producing ROS and prevents resistance mechanisms.

Some papers like J Lasers Med Sci 2018 Summer;9(3):154-160 discuss aPDT examples of biofilm eradication with respect to Staphylococcus aureus biofilms.

Specific examples related to Pseudomonas aeruginosa

Journal of Controlled Release 117 (2007) 217–226 discusses relevant to the current topic, “use of photodynamic therapy for the treatment of Pseudomonas aeruginosa cystic fibrosis pulmonary infection” and the results describe biofilms.

https://www.frontiersin.org/articles/10.3389/fmicb.2018.01299/full

https://onlinelibrary.wiley.com/doi/abs/10.1111/php.12331

https://www.hindawi.com/journals/jnm/2016/4752894/

If interested authors could comment on/include this therapeutic approach.

Overall, I found the review an easy read, good figures make it very informative and comprehensive, and will be helpful for updating current research in Pseudomonas aeruginosa biofilm eradication.

Author Response

Reviewer 1

The current review entitled “Pseudomonas aeruginosa biofilms” discusses the opportunistic human pathogen Pseudomonas aeruginosa which can cause infections in individuals with compromised immune systems. Being a nosocomial pathogen, it has the ability to form antibiotic-resistant biofilms specifically in CF of lungs. This review discusses the various aspects of Pseudomonas aeruginosa biofilms during the development stages, molecular mechanisms for invasion and persistence, diagnostics, and eradication therapies utilized.

  1. In the section discussing the biofilm composition, I was curious to know the nature of Psl exopolysaccharide, whether it is positively or negatively charged? Pel and Alginate are mentioned as cationic and anionic, respectively.

Our response: We have added the information on the charge of Psl in the revised manuscript as follows:

“Psl is a neutral pentasaccharide typically comprising of d-glucose, d-mannose and l-rhamnose moieties.”

  1. In the quorum sensing section, check ref- (Lee et al. 2013), appears twice as ref no. 93 and 95 and as (Lee et al. 2013) format.

Our response: Ref (Lee et al. 2013) has been deleted, and Refs [93, 95] that cited the same ref (Lee et al. 2013) have been revised to ref [92].

  1. Table 1 describes the summary of therapeutic strategies against  aeruginosainfections which need reformatting. The current format made it difficult to comprehend initially.

Our response: To make it easier to comprehend, Table 1 has been reformatted to standard black lines without the shading.

  1. I found that the therapeutic approaches section doesn’t discuss, examples of antimicrobial peptides, antibiotics, lectin inhibitors, bacteriophages, and natural products except in table 1 and only focuses on the remaining like nanoparticles, targeting EPS components, immunotherapies, biofilm dispersal, inhibition of QS, and iron chelation. Any specific reason for this distribution should be commented on.

Our response: On the Section 5 “Therapeutic Strategies”, we focus on discussing the targeting mechanisms of therapeutic agents, rather than discussing each of the therapeutic agents themselves. Given the rapid developments of the therapeutic agents and their vast availability and diversity, we attempted to present them as Table 1 instead. Table 1 summarizes the advantages and the limitation of the individual therapeutic agents along with corresponding references.

  1. In the introduction, authors have mentioned: “the increasing incidence of acute and persisting infections worldwide also highlights the need to develop therapeutic strategies as an alternative to traditional antibiotics, expectedly to disarm and eradicate this Gram-negative bacterium.” Although, the above-mentioned therapeutic approaches discussed are alternative, can antimicrobial photodynamic inactivation/therapy (aPDI/T) count as an alternative therapy?

For further reading please refer to-

  • General aPDI reading- 1. Critical Reviews in Microbiology, 2018, 44:5, 571-589, (DOI:10.1080/1040841X.2018.1467876).
  • Considering that the structure of biofilms is mainly composed of exopolysaccharides, (depending on their charges as described in the biofilm composition section), the aPDT therapeutic method targets cell wall structures of bacteria by producing ROS and prevents resistance mechanisms.
  • Some papers like J Lasers Med Sci 2018 Summer;9(3):154-160 discuss aPDT examples of biofilm eradication with respect to Staphylococcus aureus

Specific examples related to Pseudomonas aeruginosa

  • Journal of Controlled Release 117 (2007) 217–226 discusses relevant to the current topic, “use of photodynamic therapy for the treatment of Pseudomonas aeruginosacystic fibrosis pulmonary infection” and the results describe biofilms.
  • https://www.frontiersin.org/articles/10.3389/fmicb.2018.01299/full
  • https://onlinelibrary.wiley.com/doi/abs/10.1111/php.12331
  • https://www.hindawi.com/journals/jnm/2016/4752894/

If interested authors could comment on/include this therapeutic approach.

Our response: Thank you for the Reviewer’s comment and suggestion. We have added new subsections 5.7 and 5.8 to discuss photodynamic and photothermal therapies, respectively, and include the corresponding references. Please see lines 493-515 for Section 5.7 and lines 517-536 for Section 5.8 on the revised manuscript.

  1. Overall, I found the review an easy read, good figures make it very informative and comprehensive, and will be helpful for updating current research in Pseudomonas aeruginosabiofilm eradication.

Our response: We thank the reviewer for the positive comments of our work.

Reviewer 2 Report

Dear Authors,

I found your manuscript strongly interesting and wery well packaged.

I only suggest you to use shorter sentences in English.

Thank you for your contribute.

Author Response

Reviewer 2

Dear Authors, I found your manuscript strongly interesting and very well packaged. I only suggest you to use shorter sentences in English. Thank you for your contribute.

Our response: We thank the reviewer for the positive evaluation of our work. We have revised the long sentence in line 82 page 3 as a couple of short sentences as follows:

“Psl is a neutral pentasaccharide typically comprising d-glucose, d-mannose and l-rhamnose moieties. This exopolysaccharide is necessary for adhesion of sessile cells (cells attached to a surface) to surfaces and cell-to-cell interactions during biofilm initiation of both nonmucoid and mucoid strains.”

Reviewer 3 Report

The work was written about biofilm formation by Pseudomonas aeruginosa. It exhaustively describes the formation of biofilm and its stages. The subject of the work is interesting, not innovative, but still relevant. There are quite a lot more satirical items in the literature, from more than 10 years ago, which indicates that the topic has been largely exhausted and there is no novelty in it. Taking into account the subject matter and scoring of the journal, it should be changed in possible publication. Therapeutic strategies should be better described, that will be the novelty aspect.

The figures in the work are clear and describe the phenomenon well. The table is unreadable in this location, content is missing.

There are some mistakes e.g. lines:

  • 17 - there is no need to write full name of bacteria;
  • 16 - Streptococcus shouls be italics
  • 203 - reference has no number.

Author Response

Reviewer 3

  1. The work was written about biofilm formation by Pseudomonas aeruginosa. It exhaustively describes the formation of biofilm and its stages. The subject of the work is interesting, not innovative, but still relevant. There are quite a lot more satirical items in the literature, from more than 10 years ago, which indicates that the topic has been largely exhausted and there is no novelty in it. Taking into account the subject matter and scoring of the journal, it should be changed in possible publication. Therapeutic strategies should be better described, that will be the novelty aspect.

Our response: We thank the reviewer for the evaluation of our work and the constructive suggestion to improve this manuscript. As suggested, we have added more discussion on the therapeutic strategies to emphasize on the novelty aspect of this manuscript. The two novel photodynamic and photothermal therapies have been added in Sections 5.7 and 5.8, respectively.

  1. The figures in the work are clear and describe the phenomenon well. The table is unreadable in this location, content is missing.

Our response: The table has been reformatted. Please see page 12 on the revised manuscript.

  1. There are some mistakes e.g. lines:

  • 17 - there is no need to write full name of bacteria.

Our response: The name of the bacteria Pseudomonas aeruginosa has been revised in abbreviation version as P. aeruginosa.

  • 16 - Streptococcus shouls be italics

Our response: Streptococcus has been italicized to Streptococcus.

  • 203 - reference has no number.

Our response: The reference has been revised as ref [75]. Please see line 213 on the revised manuscript.

Round 2

Reviewer 3 Report

Despite the fact that the work does not concern an innovative topic, it comprehensively describes the issue. The manuscript is written in the correct scientific language. The authors took into account the previous comments. The table is readable and understandable, brings a lot to work about Pseudomonas aeruginosa.